# Toxicity of TiO_2_ Nanoparticles: Validation of Alternative Models

**DOI:** 10.3390/ijms21144855

**Published:** 2020-07-09

**Authors:** Mélanie M. Leroux, Zahra Doumandji, Laetitia Chézeau, Laurent Gaté, Sara Nahle, Romain Hocquel, Vadim Zhernovkov, Sylvie Migot, Jafar Ghanbaja, Céline Bonnet, Raphaël Schneider, Bertrand H. Rihn, Luc Ferrari, Olivier Joubert

**Affiliations:** 1Institut Jean Lamour, UMR CNRS 7198, Université de Lorraine, CNRS, IJL, F-54000 Nancy, France; melanie.lovera-leroux@univ-lorraine.fr (M.M.L.); doumandji.zahra@gmail.com (Z.D.); sara.nahle@univ-lorraine.fr (S.N.); romain.hocquel@univ-lorraine.fr (R.H.); syvlie.migot@univ-lorraine.Fr (S.M.); jafar.ghanbaja@univ-lorraine.fr (J.G.); bertrand.rihn@univ-lorraine.fr (B.H.R.); luc.ferrari@univ-lorraine.fr (L.F.); 2Institut National de Recherche et de Sécurité, rue du Morvan, 54519 Vandœuvre-les-Nancy, France; laetitia.chezeau@hotmail.fr (L.C.); laurent.gate@inrs.fr (L.G.); 3Systems Biology Ireland, University College Dublin, Dublin 4, Ireland; vadim.zhernovkov@ucd.ie; 4Université de Lorraine, CHRU-Nancy, Genetic Department, F-54000 Nancy, France; c.bonnet@univ-lorraine.fr; 5Laboratoire Réactions et Génie des Procédés, Université de Lorraine, CNRS, LRGP, F-54000 Nancy, France; raphael.schneider@univ-lorraine.FR

**Keywords:** titanium dioxide, nanoparticles, transcriptomics, rat, macrophages, ALI, toxicogenomics

## Abstract

There are many studies concerning titanium dioxide (TiO_2_) nanoparticles (NP) toxicity. Nevertheless, there are few publications comparing *in vitro* and *in vivo* exposure, and even less comparing air–liquid interface exposure (ALI) with other *in vitro* and *in vivo* exposures. The identification and validation of common markers under different exposure conditions are relevant for the development of smart and quick nanotoxicity tests. In this work, cell viability was assessed *in vitro* by WST-1 and LDH assays after the exposure of NR8383 cells to TiO_2_ NP sample. To evaluate *in vitro* gene expression profile, NR8383 cells were exposed to TiO_2_ NP during 4 h at 3 cm^2^ of TiO_2_ NP/cm^2^ of cells or 19 μg/mL, in two settings—submerged cultures and ALI. For the *in vivo* study, Fischer 344 rats were exposed by inhalation to a nanostructured aerosol at a concentration of 10 mg/m^3^, 6 h/day, 5 days/week for 4 weeks. This was followed immediately by gene expression analysis. The results showed a low cytotoxic potential of TiO_2_ NP on NR8383 cells. Despite the absence of toxicity at the doses studied, the different exposures to TiO_2_ NP induce 18 common differentially expressed genes (DEG) which are involved in mitosis regulation, cell proliferation and apoptosis and inflammation transport of membrane proteins. Among these genes, we noticed the upregulation of *Ccl4*, *Osm*, *Ccl7* and *Bcl3* genes which could be suggested as early response biomarkers after exposure to TiO_2_ NP. On the other hand, the comparison of the three models helped us to validate the alternative ones, namely submerged and ALI approaches.

## 1. Introduction

Among the nanomaterials used in industries, titanium dioxide (TiO_2_) nanoparticles (NP) are one of the most used. TiO_2_ NP are mainly used as pigments for their brightness, high refractive index, opacity and antimicrobial properties. They are also useful in many cosmetics applications such as in makeup, sunscreen, toothpastes and personal care products [1,2]. In the medical domain, TiO_2_ NP are used as components for prosthetic implants (hip, knees, dental implants) or in intravenous injection [3,4,5]. TiO_2_ NP are also found in other various applications like paint, glass, electronic and water treatment industries [6,7,8,9,10].

Due to these various industrial uses, TiO_2_ represent 70% of the total production volume of pigments worldwide and it is in the top five NPs used in consumer products and approximately four million tons of TiO_2_ are produced annually worldwide [4]. Their wide use must be challenged for the potential adverse health effects they can induce. Thus, it is urgent to assess the risk of different exposures to TiO_2_ nanoparticles. Indeed, the interaction of nanoparticles with living organisms could result in biologic damages. Many studies on pulmonary toxicity of TiO_2_ NPs have been published so far and *in vivo* and *in vitro* data indicate that the main toxicity mechanisms induced by TiO_2_ NPs include pulmonary inflammation and oxidative stress, as well as genotoxicity [4,11,12,13,14]. However, discrepant results about the genotoxicity of these nanomaterials could be found in the literature. Also, different *in vivo* studies showed an increased significant inflammation after ingestion of TiO_2_ NP [15,16].

Considering this bulk of studies about TiO_2_ nanoparticles (NP) toxicity, there are few publications comparing *in vitro* and *in vivo* exposures—and even fewer comparing air–liquid interface exposure (ALI) with other *in vitro* and *in vivo* exposures [17,18,19]. Therefore, the aim of this study was to identify markers of exposure of the airways to TiO_2_ NP, by comparing classical submerged *in vitro*, ALI and *in vivo* exposures.

NR8383 rat lung macrophages are relevant due to their immune functions and it is a validated model for nanotoxicological studies [20,21]. In the present work, we used this cell line with two types of exposure (submerged and ALI) compared to an *in vivo* rat exposure [22] to validate alternative models for nanotoxicological studies.

The present study succeeded in establishing a correlation of deregulated genes, which can be considered as biomarkers of exposure to TiO_2_ NPs and their associated molecular pathways on three different *in vivo* and *in vitro* models.

Here, after analyzing transcriptomes of the different models following three type of exposures to TiO_2_ NP, we identified common genes and biologic pathways. This validates alternative models, which are cost and time effective and ethically more acceptable.

## 2. Results

### 2.1. TiO_2_ Nanoparticles Characterization

The morphology of TiO_2_ NP was observed under a transmission electron microscope (TEM). The size of the NM105 nanoparticles was 21.5 ± 7.2 nm (Figure 1) which is in accordance with the characteristics given by the supplier (Table 1). X-ray diffraction analysis showed that the NM-105 NP samples were composed of about 18% rutile and 82% anatase. The secondary size obtained by DLS was around 170 nm, the zeta potential was 11.1 ± 0.7 mV and the specific surface area was 51 m^2^/g (Table 1).

### 2.2. In Vitro Cytotoxicity Study

Regardless of the tests and the doses (Figure 2), exposure of NR8383 cells to TiO_2_ NP for 24 h did not statistically decrease their viability. Indeed, compared to cells not exposed to NM-105 NP, the metabolic activity measured by the WST-1 test was constant and the 10% decrease observed after a TiO_2_ exposure of 100 µg/mL and 200 mg/mL was not statistically significant (Figure 2A). Nevertheless, the release of LDH into the extracellular medium increased up to 30 % in a statistically insignificant manner for 100-µg/mL and 200-mg/mL TiO_2_ exposure (Figure 2B).

### 2.3. Transcriptomic Analysis of Dysregulated Genes Following In Vivo, In Vitro Submerged, and ALI Vitrocell Cloud ^®^ Exposure (ALI) to NM-105 TiO_2_ NP

As represented by the volcano plot of dysregulated genes (Figure 3), ALI rat cells and lung exposed to TiO_2_ NP by inhalation showed similar number of differentially expressed genes (DEGs), 851 and 1477, respectively, with a fold change > 1.3 (Table 2), while the number of genes dysregulated of *in vitro* submerged conditions with a fold change > 1.3 was ten times higher with 9836 DEGs (Table 2, Figure 3A). With a fold change > 3, we found 1721 genes differentially expressed genes in the *in vitro* submerged cells exposed to NM-105, but only 69 DEGs in *in vivo*. Interestingly, with a fold change of 3, a higher proportion of downregulated genes were found with *in vitro* submerged condition compared to *in vivo* exposure which showed more upregulated genes than downregulated (Table 2). Finally, it is important to note that the transcriptomic study brings out only one DEG with a fold change > 3 in the case of ALI exposure, namely *Myc* (Figure 3C). Indeed, ALI exposure showed less DEGs genes with only 202 DEG with a fold change > 1.5 vs. 780 DEGs for *in vivo* exposure and 7895 for *in vitro* submerged exposure (Table 2).

The gene spring analysis of the 20 most up-DEG and 20 down-DEG in the three conditions (Appendix A) allows to identify five KEGG pathways for *in vitro* submerged cells exposed to TiO_2_ NP, including ‘IL-17 signaling pathway’, ‘Chemokine signaling pathway’ and ‘Cytokine–cytokine receptor interaction’: rno04657, rno0462, rno05323, rno04060, rno05144 (Appendix A), emphasize the role of macrophages inflammation. No common pathway was identified in the analysis of *in vivo* and ALI DEGs. Interestingly, GO biologic process (BP) annotation showed 5 common biologic process between *in vivo* and ALI: 0080090, 0060255, 051172, 0051171, 0050896 (Appendix A); all are involved in metabolic and cellular processes, while only one is common between *in vivo* and *in vitro* submerged condition, namely 0048545 (Appendix A). No common BP for ALI and *in vitro* submerged conditions or for the three conditions were evidenced. More connections between the DEGs were found *in vitro* (22) and *in vivo* (16) than in ALI conditions (6), and for all the groups, the software indicated that the genes are at least partially biologically connected as a group (Appendix A).

### 2.4. Common Dysregulated Genes between In Vivo, In Vitro, and ALI Exposures to TiO_2_ NP

Regarding the differences between the fold changes of *in vitro* submerged experiments, the results comparison was performed using a cutoff of 2.8 for *in vitro* submerged and 1.3 for both *in vivo* and *in vitro* ALI conditions with a *p*-value fixed to *p* < 0.05 for the three conditions. These cutoffs focus on the most deregulated genes in the three conditions and allow a better selective analysis for the comparison of the three exposures.

The Venn diagram of differentially dysregulated genes comparing TiO_2_ NP exposition in the three conditions showed 18 common DEGs. 107 common DEGs were found between *in vivo* (lung) and *in vitro* submerged, 53 between *in vivo* (lung) and ALI exposures and 147 between *in vitro* submerged and ALI exposures (Figure 4).

### 2.5. Comparison of Functional Annotations of Dysregulated Genes In Vivo, In Vitro, and ALI after Exposition to TiO_2_ NP

GSEA analysis showed two common gene sets dysregulated in the three exposure, related to inflammation and oncogenesis: the IL6-JAK-STAT3 signaling and genes regulated by the transcription factor MYC (MYC_TARGETS_V2) (Table 3). Four common gene sets were found between *in vivo* and ALI exposure, among them genes implied in ‘cell cycle regulation’ (E2F_TARGETS, G2M_CHECKPOINT) and ‘cell transformation’ (EPITHELIAL_ MESENCHYMAL_TRANSITION). Interestingly, another group of genes regulated by the transcription factor MYC was found (MYC_TARGETS_V1), which can be clustered in a MYC_TARGETS group V1 and V2. Two common gene sets were found between *in vivo* and *in vitro* submerged exposures (UV_RESPONSE_DN and TNFA_SIGNALING_VIA_NFKB) and no common gene set was found between *in vitro* and in ALI conditions. Seventeen gene sets were found only in lung, which is not surprising because herein the cell types analyzed are heterogenous, notably composed of epithelial, immune and endothelial cells, whereas cultured cells are characterized by their homogeneity. These gene sets were related to inflammation, immune response and homeostasis. ALI exposure resulted in UV_RESPONSE_UP gene set which can be brought closer to those common between *in vivo* and *in vitro* submerged exposures, in a UV-RESPONSE dysregulation. Finally, three gene sets were found specifically for *in vitro* submerged samples (MITOTIC_SPINDLE, DNA_REPAIR and PROTEIN_SECRETION).

### 2.6. Functional Analysis of Common Dysregulated Genes between In Vivo, In Vitro, and ALI Expositions to TiO_2_ NP

Among the 18 common DEGs identified with the Venn diagram (Figure 4), four are upregulated in the three conditions: *Ccl4* (FC: 1.71; 3.72; 2.15 for *in vivo*, *in vitro* and ALI conditions, respectively), *Ccl7* (FC: 5.61; 3.53; 1.47) and *Osm* (FC: 1.38; 6.84; 1.41) which are involved in the same KEGG pathways (cytokine–cytokine receptor interaction) and *Bcl3* (FC: 1.69; 4.60; 1.41) which plays a role in cell proliferation (Table 4 and Table 5B), while the others common DEGs displayed opposite up and down regulation. Indeed, 11 DEG are upregulated *in vivo* and downregulated *in vitro* and in ALI conditions. *Ppp2r5b* and *Osgin1* are upregulated *in vitro* and downregulated *in vivo* and *Ptpn13* is upregulated during *in vitro* submerged conditions and downregulated for both *in vivo* and ALI conditions (Figure 5). These differences can be explained easily by the experimental conditions as *in vivo* results are for the whole lung rat while *in vitro* and ALI conditions only for macrophages cell NR8383.

The 18 dysregulated genes common for the 3 exposure methods (Table 4, Figure 5) are involved in ‘mitosis regulation’, and evidenced by changes in (*i*) constitution of the kinetochore–centromere complex: *Cenpf*, *Nuf2*, *Nuf2*, *Kif15*, *Kif20b*, *Plk4*, (*ii*) mitosis regulation: *Depdc1* and (*iii*) chromatin remodeling: *Hmgb2*. Moreover significant changes were also evidenced in (*i*) ‘cell proliferation and apoptosis’ (*Bcl3*, *Osgin1*, *Ptpn13*, *Ppp2r5*, *Bcap29*), (*ii*) ‘cell differentiation’ (*Hmgb2*, *P2ry12*, *Tfec*, *Ogfrl1*, *Bcl3*, *Ptpn13*), (*iii*) ‘inflammation’ (*P2ry12*, *Ccl4*, *Ccl7*, *Osm*, *Tfec*, *Osgin1*) and (*iv*) a single gene related to transport of membrane proteins (*Bcap29*). KEGG pathway analysis found one common pathway for 3 genes (*Osm*, *Ccl4* and *Ccl7*), namely ‘cytokine–cytokine receptor interaction pathway’ that is involved in intracellular regulation and immune response, inflammation, cell growth, differentiation, cell death, angiogenesis, development and repair processes aimed at the restoration of homeostasis (Table 4, Figure 6). Unsurprisingly, several studies and databases showed that these 18 dysregulated genes are associated with tumorigenesis, cancer initiation progression and aggressiveness (Table 4).

GO-BP analysis of the 18 common DEGs between the three exposure methods highlight some groups functions: inflammation and immune response, cell migration, intracellular movements (Table 5A). Coherently, a KEGG analysis finds one pathway associated with inflammation: cytokine–cytokine receptor interaction pathway (RNO-04060) and Reactome analysis revealed five pathways implied in mitosis and transport: mitotic prometaphase (R-RNO-68877), separation of sister chromatids (RNO-2500257, RNO-2467813), kinetochore and actin functions (RNO-141444 and RNO-5663220) (Table 5B,C). When projected to human model, the same dysregulated pathways were found (Appendix A).

## 3. Discussion

### 3.1. Methodology

There are many studies about TiO_2_ nanoparticles (NP) toxicity. Nevertheless, there are few publications comparing *in vitro* and *in vivo* exposures and even less comparing air–liquid interface exposure with other *in vitro* and *in vivo* exposures. Moreover, the Vitrocell Cloud^®^ system is an innovative system using cloud exposure by a nebulizer, which mimics the lung interface. In this original study we compare the classic *in vitro* method with Vitrocell Cloud^®^ and *in vivo* exposures to TiO_2_ nanoparticles in a complete transcriptomic study. In the current state of our knowledge, it is the first publication comparing these three expositions methods. Identification of common markers of exposure or effect is relevant for the development of smart and quick tests of nanotoxicity. In addition, in order to respect the three “R” of the ethical approach outlined by Russel and Burch in 1959 to reduce, replace and refine the use of animal testing, it seems relevant to develop *in vitro* models [37]. It is obvious that the *in vitro* study of different cell lines represents a promising tool for the implementation of predictive devices for NP exposure [38]. Thus, the aim of this study was to identify and validate early specific markers of lung exposure to TiO_2_ NP, by comparing *in vitro*, air–liquid interface (ALI) and *in vivo* exposures.

Knowing that nanoparticles can reach the alveoli [39], NR8383 cells are an appropriate model because they are alveolar macrophage precursors, which are the first implicated cells in the alveolar clearance of nanoparticles [40,41,42].

This cell line has already been studied and validated as a model in the field of nanotoxicology [20,43]. Indeed, NR8383 cells are relevant for their immune functions [21]. NR8383 cells were exposed under classic submerged conditions or through an ALI cloud device (Vitrocell Cloud System^®^), to 3 cm^2^/cm^2^ (19 µg/mL of TiO_2_ NP) during 4 h.

To limit the bias due to agglomeration of nanoparticles, TiO_2_ NP were sonicated and vortexed before each treatment. However, typical submerged exposition methods do not take into account the cellular interactions, the role of alveolar surfactant, the pulmonary clearance and the differential deposition in the respiratory tract regarding the displacement of NPs in the air during the respiration [39,44,45]. Moreover, the surfactant plays an important role in the uptake of nanoparticles (Geiser, 2010). Therefore, we chose to compare classic *in vitro* method with ALI exposure using the same cell line, exposed with a nebulizer first to the surfactant and then to TiO_2_ NP, at the same dose.

The *in vivo* study was carried out on Fischer 344 rats exposed by inhalation (nose only) of 10 mg/m^3^, 6 h/day, 5 days/week for 4 weeks to TiO_2_ NP NM-105. Immediately after the last exposure of the rats, the genes expression was analyzed by transcriptomics and compared to *in vitro* results on NR8383 rat macrophages submerges and in air–liquid interface exposure.

### 3.2. Viability of NR8383 Cells Exposed to TiO_2_

NR8383 cells were exposed to TiO_2_ NP from 0.25 to 200 µg/mL for 24 h. The viability tests showed a little, although not significant, decrease of mitochondrial activity activity up to 10 % decrease for 100 and 200 µg/mL of NM105 in WST-1 test and for cell membrane integrity up to 30 % decrease for 100 and 200 µg/mL of NM105 in LDH test (Figure 2). The real cytotoxic potential of TiO_2_ NP remains uncertain and depends on their size, agglomeration and crystalline composition: anatase seems to be less toxic than rutile form [46,47]. In addition, a review reports no reduction in viability for BEAS-2B cells exposed to NM-105 at 150 µg/mL [48]. Another study showed no loss of cell membrane integrity in a 3D human bronchial model exposed to TiO_2_ NP [49]. At the opposite, Park et al. found a significant loss of 30 % viability, by MTT assay, in BEAS-2B cells exposed to 40 µg/mL of TiO_2_ NP while Wiemann et al. found a significant loss of cell membrane integrity, by LDH assay, in NR8383 exposed to 90 and 180 µg/mL of NM-105 TiO_2_ NP: 53.6 % and 69.2 %, respectively [50]. A high toxicity of TiO_2_ NP was also found in RLE-6TN rat alveolar epithelial cells: IC50 was found to be 7 µg/mL following 24 h exposure [51].

### 3.3. Number of DEGs in Each Exposure Method

The number of differentially expressed genes was higher in *in vitro* submerged model than the others regarding both the number of genes and fold changes (1721 DEGs with a fold change > 3, Table 2), whereas ALI exposure resulted in less differences between exposed and control cells (1 DEG with a fold change > 3, Table 2). Given a fold change > 1.3, the number of genes dysregulated for *in vitro* submerged cells is ten times bigger than the number of genes dysregulated *in vivo* and in ALI conditions: 9836, 851 and 1477, respectively (Table 2). These differences are probably due to the exposition mode and to the cell type model. Indeed, *in vitro* submerged exposure is the common method used in toxicology studies, but it showed some limits: the sedimentation of nanoparticles is heterogenous because of the aqueous behavior, agglomeration, interaction with the solvent and proteins, density, dispersion, dissolution and sedimentation [52,53,54]. Consequently, the cells can be heterogeneously exposed with some cells exposed in excess and some not exposed at all [53]. High numbers of DEG genes between for *in vitro* submerged cells was already observed in previous transcriptomics studies [55,56,57]. This is why we chose to focus on DEGs with a FC > 2.8 for *in vitro* submerged exposition.

ALI NR8383 cell culture with surfactant have showed less DEG genes than classic *in vitro* exposed cells without surfactant, this may be explained by the short exposure time and the surfactant layer above cells. Nevertheless, some studies showed that surfactant allows a better uptake of TiO_2_ nanoparticles by alveolar macrophages. Pulmonary surfactant is composed of phospholipids and lipids (80% and 10%, respectively) and 10% proteins, among them, collectin proteins (SP-A, B, C, D) which rapidly opsonize the particles and modulate the immune response by increasing the macrophages phagocytosis [58,59]. This may suggest that the ALI exposition method with surfactant before TiO_2_ exposure should have improved the uptake and the cell response and consequently the number of DEG compared to classic *in vitro* exposure without surfactant. But this is not the case in our study, and this implies that other mechanisms could play a role. Opsonization by surfactant proteins is a rapid process that occurs between 5 and 15 min [60], so the time course of opsonization does not seem to be involved in our results.

However, dispersion and settling of NP depend on the pH of surfactant, its composition and its concentration [61,62]. TiO_2_ dispersion stability and agglomeration depend also on the electronic properties of the surfactant [63,64]. The formation of micelles could lead to the engulfment of NP [65,66,67] and TiO_2_ NPs are more electronegative and agglomerate less in electrolytic surfactants [68,69]. Indeed, TiO_2_ NP dispersion was shown to remain stable in surfactant, but NPs settles easily at the bottom in media without surfactant [61], probably reducing the contact of NPs with cells. As a confirmation, this hypothesis was already observed with other NPs, for example, cells treated four hours with SiO_2_ NP in the presence of pulmonary surfactant have showed less internalization of NPs, and microscopic observations of the cells showed NPs with less agglomeration and floating above the cell culture [70].

To elucidate the influence of surfactant in macrophages response to TiO_2_ NPs, it will be crucial to further study the composition of rat surfactant, the turbidity of this surfactant with TiO_2_, the deposition time of TiO_2_, and electronic microscopy imaging of NR8383 cells with surfactant exposed to NP TiO_2_ NM-105 at different time points.

TiO_2_ NP was a mixed phase nanocrystalline powder composed of approximately 80% anatase and 20% rutile, with an average primary particle diameter of 21.5 ± 7.2 nm (Table 1). TiO_2_ aerosol was generated at a target concentration of 10 mg/m^3^. Taking into account the interspecies differences in terms of respiratory parameters and lung deposited dose, this concentration and exposure time relate to the 8 h weighted average occupational exposure of a worker at 0.3 mg/m^3^ during most of its career, the NIOSH recommended exposure limit for ultrafine TiO_2_ [71,72].

In a concomitant study realized with F344 rats, at the end of the inhalation period, the concentration of titanium in lung tissue was approximately 2 mg/lung in exposed rats [73]. In the lung, after inhalation, the particles are preferentially deposited in the proximal alveolar region (PAR). Then, the surface area of particles per unit area of PAR is approximately 5 cm^2^/cm^2^ [73] while the threshold dose for the onset of inflammation is estimated at approximately 1 cm^2^/cm^2^ [74]. The presence of a strong pulmonary inflammation at the end of the exposure, agrees with these data. Finally, the presence of macrophages loaded with TiO_2_ NP in the alveoli and the lymph nodes associated with the lungs [22], suggests a conservation of the mechanisms of particle elimination, by the mucociliary escalator or by translocation from the alveolar region to the lymph nodes. These results demonstrate the key role of alveolar macrophages in the pulmonary response to nanoparticle inhalation and justify the comparison of *in vivo* response with that of cultured rat alveolar macrophages.

### 3.4. Transcriptomic Study

The analysis of the 20 most up-DEG and 20 most down-DEG in the three conditions highlights five KEGG pathways dysregulated for *in vitro* submerged cells exposed to TiO_2_ NP including ‘IL-17 signaling pathway’, ‘chemokine signaling pathway’ and ‘cytokine–cytokine receptor interaction’. Interestingly, ‘Il-17 signaling pathway’ is commonly associated with the activity of T lymphocytes, but macrophages were already demonstrated to be involved in Il-17 mediated inflammation as suggested by the (*i*) upregulation of IL17 receptors in vitro, (*ii*) their production of unique profiles of cytokines and chemokines [75] and (*iii*) the activation of macrophages in a M1/M2 heterogeneous phenotype [76]. Five common biologic process (GO-BP) were found between *in vivo* and ALI exposures, only one between *in vivo* and *in vitro* exposures and none between ALI and *in vitro* exposures. The 5 common GO-BP are involved in metabolic and cellular processes, suggesting that cellular responses in ALI conditions are similar to those obtained for *in vivo* condition.

GSEA analysis highlighted two common gene sets dysregulated in all three exposure, related to inflammation and oncogenesis: the IL6-JAK-STAT3 signaling and genes regulated by the transcription factor MYC (MYC_TARGETS_V2) (Table 3). IL6-JAK-STAT3 pathway is known to be implied in tumorigenesis [77] and play a role in cancer-associated inflammatory environment [78] and angiogenesis [79]. Genes regulated by the transcription factor *Myc* are mainly involved in cell growth, apoptosis and metabolism. The *Myc* gene is well known as a proto-oncogene that is over expressed in various types of cancers include lymphomas, lung carcinoma, breast carcinoma and colon carcinomas [80]. *Myc* is also known to be involved in the polarization of M2 tumor-associated macrophages and is targeted for new therapeutic strategies against cancer [81,82,83,84].

Interestingly, as opposed to what could be expected, *Myc* was the most downregulated gene in ALI exposures (Table 2). However, the role of this transcription factor is large and remain poorly understood. Some studies have shown that downregulation of *Myc* gene can induce cell cycle arrest and, in some case cell survival, differentiation or apoptosis [85,86,87]. As no cell death was observed after NM105 TiO_2_ NP exposure in our study, we can suppose that the down regulation of *Myc* in NR8383 exposed in ALI experiments, may be associated with a cell cycle arrest and survival of NR8383. Cell survival and cell cycle states in these conditions will be further investigated.

To propose biomarkers of effect for NM-105 TiO_2_, we focused our analyses on the common dysregulated genes (DEGs) between the three exposure modes. 18 common DEGs were found, among them, only 4 are upregulated in the three conditions: *Ccl4*, Ccl7, *Osm* and *Bcl3*, while the other common DEG display a non-homogenous dysregulation. Indeed, among the 18 DEGs, 14 DEGs showed the same dysregulation profile following *in vitro* and ALI conditions: among them 12 DEGs were down regulated and 2 upregulated as noted in Table 4. Contrary those genes were downregulated *in vivo*. Only *Ptpn13* was downregulated *in vivo* and in ALI conditions while upregulated *in vitro* (see Table 4). These discordant results can be explained by the experiments conditions. For *in vivo* study, transcriptomic analyses consider all the cellular perturbations induced by NP exposure as well as the interaction between the different cell types including macrophages, epithelial, interstitial and endothelial cells, after a subacute exposure (6 h/day, 5 days/week for 4 weeks) while only NR8383 cells are grown for a short 4 h exposure in both vitro tests. Moreover, surfactant was present in ALI and *in vivo* conditions, but not for *in vitro* submerged conditions.

Consistently with our previous observations, common dysregulations for *in vitro* and ALI *versus*
*in vivo* exposures revealed a negative regulation of mitosis and cell proliferation and a negative regulation of M2 polarization of macrophages (Table 4). These observations are in accordance with the hypothesis of a cell cycle arrest without any cytotoxicity of macrophages *in vitro* and in ALI conditions. It can be an early effect of TiO_2_ NP on macrophages, while the upregulation of these genes for *in vivo* conditions can be due to other cell types in lung or to an adaptative regulation towards proliferation after a longer *in vivo* exposure.

Note that macrophages are ubiquitous innate immune cells and can be polarized in two distinct functional phenotypes, either in pro-inflammatory macrophages called M1 or in M2 macrophages that promote immune suppression and wound healing [88,89,90]. They also suggest a preferential differentiation in M2 macrophages *in vivo* compared to M1 that were found *in vitro* and in ALI conditions. It is well known that in cancer pathologies, M2 macrophages can be recruited and favor tumor growth by contributing to the tumor microenvironment; by the way, they are called tumor-associated macrophages (TAM). Furthermore, *Ptpn13* is a tumor suppressor gene that is frequently inactivated in non–small cell lung cancer (NSLC) (in more than 70% of NSLC) [91], and was downregulated in our ALI and *in vivo* exposure to TiO_2_ while upregulated in *in vitro* condition.

The four genes upregulated in the three exposure conditions are actor of the cytokine–cytokine receptor interaction KEGG pathway: *Ccl4*, CcL7, *Osm* and *Bcl3* is implied in cell proliferation regulation (Table 4). Interestingly, *Ccl4* was already identified as a candidate for biomarker for BCR pathway activation and prognostic in diffuse large B cell lymphoma [92] and as an inflammation-related diseases biomarker (Olink proteomics). *Ccl7* (also called *MCP3*) plays a crucial role in cancer as it can promote tumor growth, tumor microenvironment, invasion and metastasis [93]. Indeed, overexpression of *Ccl7* was associated with lung adenomas [94] and metastasis in colorectal cancer and renal carcinoma [95,96,97]. Moreover, *Ccl7* expression is associated with the recruitment of tumor-associated macrophages (TAMs) [98]. Likewise, oncostatin M (*Osm*) is identified as an inflammation biomarker in periodontal diseases [99], sepsis [100], inflammatory bowel disease [101,102]. Other studies showed that *Osm* promotes cancer cell plasticity [103] and the polarization of pro-fibrotic M2-like macrophages [104]. Finally, *Bcl3* (B-cell lymphoma 3 gene) is a proto-oncogene whose product is an inhibitor of *NF-kappa B* [105,106] and display anti-apoptotic functions. *Bcl3* is involved in some cancers, the most known is his role in lymphoma, in which he regulates pro-survival and pro-inflammatory gene expression [107]. *Bcl3* is also involved in solid tumor progression [108] like ovarian [109], cervical cancers [110,111], mammary cancer metastasis [112] and tumor progression in breast cancer [113]. In macrophages, *Bcl3* was showed to be a key mediator of IL-10 macrophage polarization to an immunosuppressive phenotype as observed in TAM [88].

## 4. Materials and Methods

### 4.1. TiO_2_ NP Characterization

TiO_2_ NP mainly composed of anatase and rutile (82:18) were obtained by the Joint Research Center (NM-105, JRC). TiO_2_ were resuspended in ultrapure water (18 mΩ) at a concentration of 2.56 mg/mL [114], then directly sonicated (Vibracell 75022, Bioblock, Illkirch-Graffenstaden, France) at a magnitude of 30%, during 6 min under a permanent cooling. After the sonication, TiO_2_ NP were diluted at a 6.25 µg/mL in DMEM-SVF free cell culture medium (Sigma-Aldrich, Saint-Louis, MO, USA). NP morphology was observed by transmission electronic microscope (TEM). The hydrodynamic diameter and the polydispersity index were measured by dynamic light diffusion (DLS, ZetasizerTM3000E, Malvern Instruments, Worcestershire, UK). The *zeta* potential was measured by Smoluchowski’s equation [115].

### 4.2. Cell Culture

NR8383 cells were obtained from the ATCC (American Type Culture Collection, Manassas, VA, USA). Cells were cultured in DMEM medium completed by 15% of fetal bovine serum (FBS), 100 U/mL of penicillin, 100 µg/mL of streptomycin, 0.25 µg/mL of amphotericin and 2 mM of L-glutamine at a 37 °C temperature and under a 5 % CO_2_ atmosphere. For all tests, the density used was of 5 × 10^4^ cells/mL.

### 4.3. In Vitro Cytotoxicity Study

Cell viability was assessed with the WST-1 assay and the measure of the lactate dehydrogenase (LDH) release in the extracellular medium. Non-exposed cells were considered as a control. Six technical and four biologic replicates were used per each condition. After a 24-h exposure to TiO_2_ NP, cells were incubated with 5% of WST-1, during 2 h at 37 °C. Then, absorbance was measured at 450 nm, with a 690 nm as reference wavelength (iMarkTM, Bio-Rad). LDH dosage was carried out following manufacturer’s recommendations. Briefly, after 24 h of exposure to TiO_2_ NP, cells were incubated 30 min with 100 µL of mix of LDH buffer and substrate, at 20 °C. Then, 50 µL of stop solution were added and the absorbance was measured at 490 nm (iMarkTM, Bio-Rad). The positive control was the result of cells exposed to the lysis buffer (Triton 5%) during 15 min before the measure. Data are presented as means ± standard deviation (SD) of the four biologic replicates. Statistical differences were determined by a one-way analysis of variance (one-way ANOVA) followed by the post hoc Dunnett’s test.

### 4.4. In Vivo Exposure

The animal experiments were performed according to European (Directive 2010/63/EC) and French (Décret n°2013-118) legislations regarding the protection of animals used for scientific purposes. The INRS animal facility has full accreditation (authorization n°D54-547-10) from the French Ministry of Agriculture. This study was approved on October 14th, 2013 by the regional ethical committee (CELMEA n°066) appointed by the Ministry of higher education and research (Authorization n°00692.01).

The *in vivo* exposure was made at the National Institute for Research and Safety (INRS) and described previously [22]. Briefly, thirteen-week-old male Fisher F344 rats (Charles River Laboratories, France) were housed in standard environmental conditions (relative humidity: 55 ± 10 %; temperature: 22 ± 2 °C and 12/ 12 h light/dark cycle). Water and a standard laboratory animal diet (A04, Safe diet) were freely available. Two weeks before exposure, rats were gradually acclimatized to the restraining tubes. Animals were then exposed by nose-only inhalation either to filtered air (controls) or TiO_2_ aerosol at a concentration of 10 mg/m^3^ for 6 h/day, 5 days/week for 4 weeks. Immediately after the exposure period, the animals were anesthetized with pentobarbital (60 mg/kg) and exsanguinated through the abdominal aorta and lung tissues were collected. Gene expression was analyzed by transcriptomic analysis on the entire pulmonary accessory lobe. We reanalyzed the transcriptomic results obtained *in vivo* (GEO Accession: GSE99997) [22] in order to compare them with the results obtained *in vitro* and in ALI conditions.

### 4.5. Exposure at the Air–liquid Interface

Cells in air–liquid interface were exposed to aerosolized NM-105 nanoparticles using the Vitrocell cloud System^®^. ALI exposure methodology used here is described in detail in another study [116]. Briefly, NR8383 were seeded in Transwell^®^ culture inserts (Polyester membrane, TC-treated, Corning, USA) with 300.000 cells/insert during 24 h before air–liquid interface exposure with the nebulization of pig surfactant and then NM-105 NP. After exposure to 3 cm^2^ of TiO_2_ NP/cm^2^ of cell culture, cells were incubated 4 h in a 37 °C, under a 5% CO_2_ atmosphere, and RNA extractions were performed.

### 4.6. Transcriptomic Study

*Total RNA Extraction.* In order to evaluate gene expression profile, total RNA was extracted from NM-105 exposed cells during 4 h at 3 cm^2^ of TiO_2_ NP/cm^2^ of cell culture (same dose for ALI and submerged experiments), using RNA-Solv (R6830-02, USA). The unexposed cells were used as a control. The RNA concentration was determined by measuring the absorption at 260 nm using a spectrophotometer (Biotech-BioSpec-Nano, Shimadzu). The optimal purity of the RNA was ensured by the determination of 260/280 nm of an absorbance ratio A260/A280 > 1.8. RNA integrity was confirmed with the Agilent 2100 and RNA 6000 Nano LabChip bioanalyzer kits (Agilent Biotechnologies, Palo Alto, CA, USA). The threshold of 8 for RNA integrity number (RIN) was chosen as a cutoff to determine whether the extracted RNA was qualified or not.

*Expression microarray hybridization.* One hundred nanograms of RNA from each sample was labeled with cyanine 3-CTP using the low input quick amp labeling kit (Agilent Technologies). The labeled cRNAs were purified and hybridized on the Agilent G4853A Sure Print G3 Rat GE 8 * 60 K microarray chips (Agilent Technologies) that cover the entire rat transcriptome. The slides were then washed and scanned using the Agilent G2505C microarray scanner with a resolution of 3 μm. The data were extracted using Feature Extraction software version 11.0 (Agilent).

*Bioinformatics analyses.* First, the data were standardized using the GeneSpring software. The Student’s test followed by the Benjamini–Hochberg correction and the filtering criteria were applied, to identify the genes whose level of expression was considerably modified. Genes with expression changes at a fold change (FC) > |1.5| compared to control with *p* < 0.05 were considered differentially expressed at a significant level. Genes were clustered into groups according to different criteria, such as the terms gene ontology (GO) biologic process and pathways (KEGG and Reactome) (see Table 3). Gene Set Enrichment Analysis (GSEA) functional annotation was done from MSigDB Collections. For biologic interpretation, we considered the GO biologic process and pathways with a *p* ≤ 0.05 value in the cluster with an enrichment score (Z-score) greater than 1.3 [117]. Venn diagram were realized thanks to GeneVenn online software [118]. The heatmap was realized with ClustVis software online [119].

## 5. Conclusions

This work suggests that *Ccl4*, *Osm*, *Ccl7* and *Bcl3*, are early response biomarkers to TiO_2_ NP. Interestingly, *Ccl7* and *Bcl3* genes were predicted as biomarkers of TiO_2_ effects using a machine learning approach and mice *in vivo* datasets by our colleagues within the SmartNanotox project (preliminary results, Vadim Zhernovkov). This analysis permits us to identify five common GO-BP between *in vivo* and ALI exposures and only one between submerged and *in vivo*, suggesting that ALI better reflects the effect of TiO_2_ NP exposure *in vivo*. It could be interesting, in a future work, to assess other nanoobjects, such as metal oxides or carbonaceous, to verify if these genes are modified too, suggesting a “nano” effect or if not, validating a specific titanium oxide response.

Thus, this work justifies the pertinence of our *in vitro* models, which, regardless of ethical considerations, are cost and time effective, as *in vivo* experiments could produce valuable results in 8 months and *in vitro* within two weeks.

## Figures and Tables

**Figure 1 ijms-21-04855-f001:**
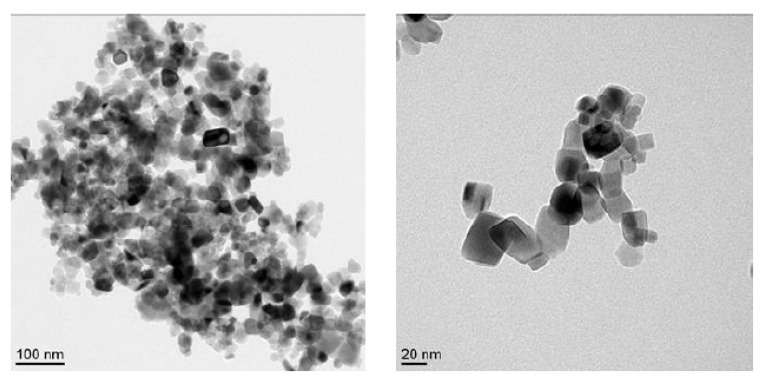
TEM images of titanium dioxide nanoparticles. TiO_2_ NPs (NM-105) in anatase dominant form.

**Figure 2 ijms-21-04855-f002:**
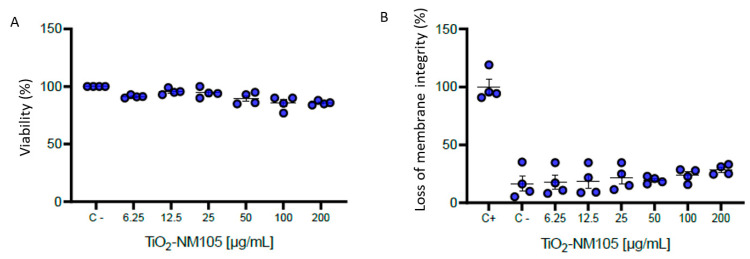
(**A**) Cytotoxicity of TiO_2_ NP (24 h exposure) to NR8383 by WST-1 test and (**B**) loss of membrane integrity by LDH release measurement. Non exposed cells are negative control (C−) and positive control (C+) for LDH are cells exposed to the lysis buffer (Triton 5%) during 15 min before the measure. Data are presented as means ± standard deviation (SD) of the four biologic replicates.

**Figure 3 ijms-21-04855-f003:**
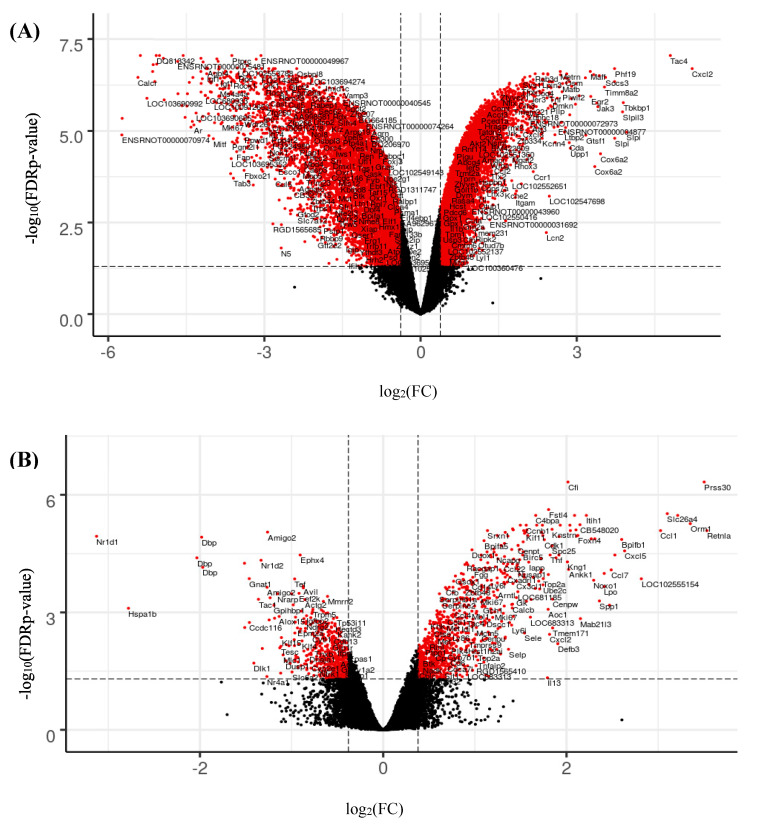
Volcano plots displaying differentially expressed genes between NM-105 treated and control samples for (**A**) *in vitro* submerged, (**B**) *in vivo* lung and (**C**) *in vitro* ALI exposures. Vertical axis corresponds to the -log_10_(FDR corrected *p*-value) and the horizontal axis displays the log2-fold change value. The vertical lines correspond to 1.3-fold up and down changes (log_2_FC = 0.38), respectively, and the horizontal line represents FDR corrected *p*-value of 0.05 (-log_10_P = 1.3). Red dots represent the up- and downregulated expressed genes.

**Figure 4 ijms-21-04855-f004:**
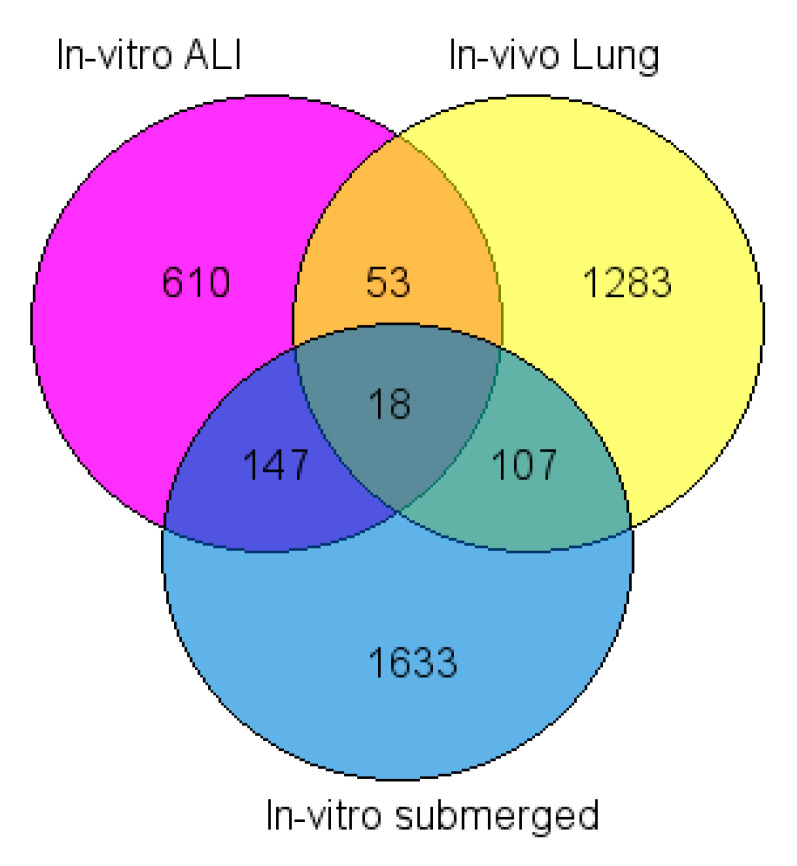
Venn diagram of differentially deregulated genes after TiO_2_ NP *in vivo* (lung), *in vitro* submerged and ALI (Vitrocell) exposures (FDR corrected *p*-value < 0.05). Genes were filtered with different fold change cutoff: *In vitro* submerged: 2.8, *In vitro* ALI Vitrocell: 1.3 and *In vivo* lung: 1.3.

**Figure 5 ijms-21-04855-f005:**
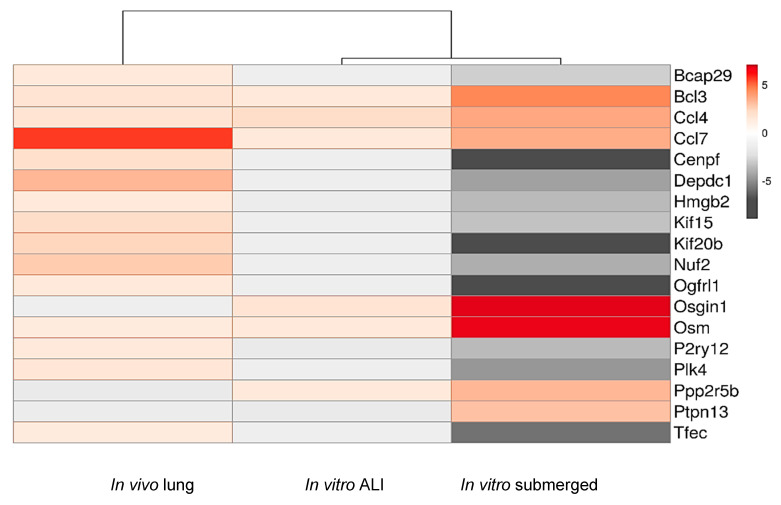
Heatmap of common dysregulated genes, fold changes are represented by a color scale (*p*-value < 0.05).

**Figure 6 ijms-21-04855-f006:**
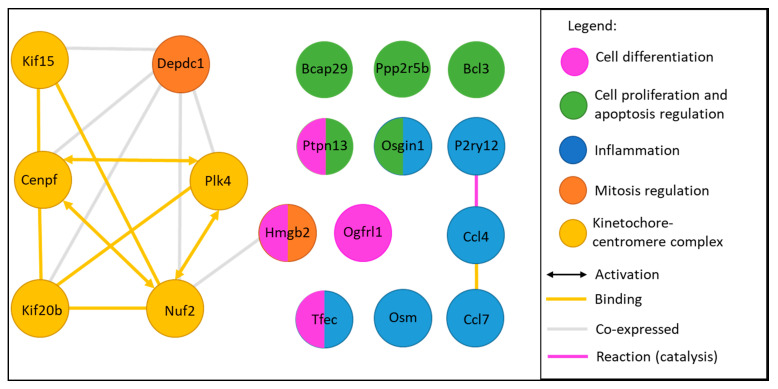
Interactions and function groups of the 18 common dysregulated genes (adapted from an analyze by String 11.1 Database, functions groups were determined through UniProt, GeneCards and PubMed researches).

**Table 1 ijms-21-04855-t001:** TiO_2_ NP characteristics.

Nanoparticle	Primary Size (nm)	Secondary Size (nm)	Zeta Potential (mV)	Specific Surface Aera (m^2^/g)	Provider
**TiO_2_ (NM-105)**	21.5 ± 7.2	170 ± 1.5	11.1 ± 0.7	51	Joint Research Center

**Table 2 ijms-21-04855-t002:** Total number of differentially expressed genes (FDR corrected *p*-value < 0.05).

Groups	FC 1.3			FC 1.5			FC3		
	Up	Down	Total	Up	Down	Total	Up	Down	Total
*in vitro* submerged	4612	5224	**9836**	3939	3956	**7895**	345	1376	**1721**
*in vitro* ALI	439	412	**851**	108	94	**202**	0	1	**1**
lung	898	579	**1477**	559	221	**780**	64	5	**69**

**Table 3 ijms-21-04855-t003:** Gene set enrichment analysis (GSEA) functional annotation of differentially expressed genes (DEG) in the three conditions.

Gene Set	*p-Value* *in vivo* Lung	*p-Value* *in vitro* ALI	*p-Value* *in vitro* Submerged
**Common gene sets between the three exposures**			
IL6_JAK_STAT3_SIGNALING	1.25 × 10^−7^	7.42 × 10^−2^	3.82 × 10^−2^
MYC_TARGETS_V2	4.30 × 10^−6^	7.42 × 10^−2^	3.65 × 10^−2^
**Common gene sets between *in vivo* and ALI exposures**			
E2F_TARGETS	1.04 × 10^−14^	8.40 × 10^−7^	–
G2M_CHECKPOINT	7.96 × 10^−11^	7.00 × 10^−3^	–
MYC_TARGETS_V1	1.08 × 10^−4^	7.00 × 10^−3^	–
EPITHELIAL_MESENCHYMAL_TRANSITION	9.03 × 10^−2^	3.84 × 10^−2^	–
**Common gene sets between *in vivo* and *in vitro* submerged exposures**			
UV_RESPONSE_DN	6.21 × 10^−3^	–	6.73 × 10^−2^
TNFA_SIGNALING_VIA_NFKB	9.43 × 10^−3^	–	1.46 × 10^−3^
**Non common gene sets**			
ALLOGRAFT_REJECTION	1.25 × 10^−7^	–	–
MTORC1_SIGNALING	5.73 × 10^−7^	–	–
INFLAMMATORY_RESPONSE	1.48 × 10^−6^	–	–
MYOGENESIS	4.30 × 10^−6^	–	–
INTERFERON_GAMMA_RESPONSE	8.61 × 10^−4^	–	–
COMPLEMENT	5.05 × 10^−3^	–	–
ANGIOGENESIS	5.82 × 10^−3^	–	–
IL2_STAT5_SIGNALING	2.32 × 10^−2^	–	–
CHOLESTEROL_HOMEOSTASIS	3.89 × 10^−2^	–	–
APICAL_JUNCTION	4.49 × 10^−2^	–	–
GLYCOLYSIS	6.27 × 10^−2^	–	–
HEDGEHOG_SIGNALING	6.27 × 10^−2^	–	–
WNT_BETA_CATENIN_SIGNALING	8.13 × 10^−2^	–	–
COAGULATION	8.89 × 10^−2^	–	–
TGF_BETA_SIGNALING	9.03 × 10^−2^	–	–
OXIDATIVE_PHOSPHORYLATION	9.03 × 10^−2^	–	–
UNFOLDED_PROTEIN_RESPONSE	9.55 × 10^−2^	–	–
UV_RESPONSE_UP	–	7.42 × 10^−2^	–
MITOTIC_SPINDLE	–	–	9.38 × 10^−3^
DNA_REPAIR	–	–	3.65 × 10^−2^
PROTEIN_SECRETION	–	–	4.11 × 10^−2^

**Table 4 ijms-21-04855-t004:** Common genes between the three exposures (different group functions are highlighted in different colors, detailed information where found in GeneCards, UniProt, String, protein Atlas and PubMed databases). Colors in the function group column are used to identify common elements easily.

Name Gene Protein	FC			Protein Function	Function Group *KEGG Pathway	Pathologies Associated
	Vivo	Vitro	ALI			
**Cenpf** **Centromere protein F**	1.97	−8.43	−1.33	The CENPF protein is a part of the corona of kinetochore complex which interacts with microtubules and participate to a precise and rapid chromosome segregation. [23]	**Mitosis** **kinetochore–centromere complex**	
**Nuf2** kinetochore protein Nuf2	2.66	−4.05	−1.31	Component of the essential kinetochore-associated NDC80 complex, required for chromosome segregation and spindle checkpoint activity, required for kinetochore integrity. [24]	**Mitosis** **kinetochore–centromere complex**	
**Kif15** kinesin-like protein KIF15	2.11	−3.35	−1.38	Plus-end directed kinesin-like motor enzyme involved in mitotic spindle assembly [25]	**Mitosis** **kinetochore–centromere complex**	
**Kif20b** kinesin family member 20B	2.50	−8.91	−1.39	Belongs to the TRAFAC class myosin–kinesin ATPase superfamily. kinesin family (String DB)	**Mitosis** **kinetochore–centromere complex**	
**Plk4** serine/threonine protein kinase	1.67	−4.64	−1.36	serine/threonine–protein kinase that plays a central role in centriole duplication; (UniProt)	**Mitosis** **kinetochore–centromere complex**	
**Depdc1** DEP domain containing 1	3.31	−4.45	−1.33	DEP domain containing 1 (DEPDC1) is a highly conserved protein among many species. DEPDC1 was overexpressed in different types of cancers. [26]	**Mitosis regulation** [26]	**Cancer** [26]
**Hmgb2** high mobility group box 2	1.46	−3.59	−1.44	Multifunctional protein with various roles in different cellular compartments. May act in a redox sensitive manner. In the nucleus is an abundant chromatin-associated non-histone protein involved in transcription, chromatin remodeling and V(D)J recombination. HMGBs act as architectural facilitators in the assembly of nucleoprotein complexes. [27]	**Mitosis** **chromatin remodeling** **T cells differentiation**	**Cancer** [28] [29]
**P2ry12** P2Y purinoceptor 12	1.43	−3.54	−1.58	Receptor for ADP and ATP coupled to G-proteins. Required for normal platelet aggregation and blood coagulation. [30]	**Inflammation: chemotaxis receptor** **Cell differentiation Macrophages M1/M2** [31]	**Cancer** [31]
**Ccl4** C–C motif chemokine 4	1.71	3.72	2.15	Monokine with inflammatory and chemokinetic properties; (UniProt)	**Inflammation: chemotaxis** *Cytokine–cytokine receptor interaction	**Inflammation diseases**
**Ccl7** C–C motif chemokine 7	5.61	3.53	1.47	Chemotactic factor attracts monocytes and eosinophils, but not neutrophils. (String DB)	**Inflammation: chemotaxis** *Cytokine–cytokine receptor interaction	**Inflammation diseases** **Cancer**
**Osm** oncostatin-M	1.38	6.84	1.41	Growth regulator. It regulates cytokine production, including IL-6, G-CSF and GM-CSF; (UniProt)	**Inflammation: regulation** *Cytokine–cytokine receptor interaction	**Cancer**
**Tfec** transcription factor EC	1.35	−5.86	−1.32	transcriptional regulator that acts as a repressor or an activator; (UniProt)	**Cell differentiation Macrophages M2 activation** [32] **Inflammation** [33]	**Cancer**
**Ogfrl1** opioid growth factor receptor-like 1	1.47	−7.00	−1.32	Mobilization and differentiation of bone marrow (BM)-derived cells [34]	**Cell differentiation ** Upregulated in M2 macrophages [35]	**Cancer** [36]
**Bcl3** B-cell CLL/lymphoma 3	1.69	4.60	1.41	BCL3 (BCL3 transcription Coactivator) is a proto-oncogene candidate. Its related pathways are Apoptosis-related network and Common cytokine receptor gamma-chain family signaling pathways. Contributes to the regulation of cell proliferation and to the regulation of transcriptional activation of NF-kappa-B target genes. (GeneCards)	**Cell proliferation** **Apoptosis regulation**	**Cancer**
**Osgin1** oxidative stress induced growth inhibitor 1	−1.38	7.10	1.74	This gene encodes an oxidative stress response protein that regulates cell death. Expression regulated by p53 and induced by DNA damage. The protein regulates apoptosis by inducing cytochrome c release from mitochondria. Key regulator of both inflammatory and anti-inflammatory molecules. The loss of this protein correlates with uncontrolled cell growth and tumor formation. (GeneCards)	**Inflammation regulation** **Apoptosis regulation**	**Cancer**
**Ptpn13** protein tyrosine phosphatase non-receptor type 13	−1.41	3.01	−1.57	Member of the protein tyrosine phosphatase (PTP) family. PTPs are signaling molecules that regulate a variety of cellular processes including cell growth, differentiation, mitotic cycle and oncogenic transformation. Regulates negatively FAS-induced apoptosis and NGFR-mediated pro-apoptotic signaling. (GeneCards)	**Cell proliferation and differentiation** **Apoptosis regulation** **Mitosis** **Oncogenic transformation**	**Amyloidosis Cancer**
**Ppp2r5b** serine/threonine protein phosphatase regulatory subunit beta isoform	−1.57	3.32	1.48	The product of this gene belongs to the phosphatase 2A (PP2A) regulatory subunit B family. PP2A it is implicated in the negative control of cell growth and division. The phosphorylated form mediates the interaction between PP2A and AKT1. (GeneCards)	**Cell proliferation**	**Cancer**
**Bcap29** B-cell receptor-associated protein 29	1.54	−2.90	−1.38	Among its related pathways are B Cell Receptor Signaling Pathway and AKT Signaling Pathway. May play a role in transport of membrane proteins. May be involved in CASP8-mediated apoptosis. (GeneCards)	**Transport of membrane protein** **Apoptosis**	

**Table 5 ijms-21-04855-t005:** (**A**) Gene ontology (GO) biologic process (20 most dysregulated GO-term); (**B**) KEGG pathways and (**C**) reactome pathways of the 18 common dysregulated genes (analyzed by String 11.1 Database).

**(A) GO-term**	**Description**	**Count in Gene Set**	**False Discovery Rate**
GO:0071346	cellular response to interferon-gamma	2 of 39	0.0265
GO:0070098	chemokine-mediated signaling pathway	2 of 30	0.0265
GO:0050921	positive regulation of chemotaxis	2 of 68	0.0265
GO:0048522	positive regulation of cellular process	7 of 2201	0.0265
GO:0048247	lymphocyte chemotaxis	2 of 20	0.0265
GO:0045087	innate immune response	3 of 217	0.0265
GO:0044089	positive regulation of cellular component biogenesis	3 of 220	0.0265
GO:0040011	locomotion	4 of 404	0.0265
GO:0030593	neutrophil chemotaxis	2 of 23	0.0265
GO:0016477	cell migration	3 of 293	0.0265
GO:0010469	regulation of signaling receptor activity	4 of 325	0.0265
GO:0009967	positive regulation of signal transduction	4 of 638	0.0265
GO:0006955	immune response	4 of 386	0.0265
GO:0006954	inflammatory response	3 of 250	0.0265
GO:0006935	chemotaxis	3 of 172	0.0265
GO:0006928	movement of cell or subcellular component	4 of 486	0.0265
GO:0002687	positive regulation of leukocyte migration	2 of 63	0.0265
GO:0002548	monocyte chemotaxis	2 of 12	0.0265
GO:0071347	cellular response to interleukin-1	2 of 75	0.0267
GO:0051173	positive regulation of nitrogen compound metabolic process	5 of 1184	0.0267

**(B) KEGG Pathways**	**Pathway Description**	**Count in Gene Set**	**False Discovery Rate**
rno04060	Cytokine–cytokine receptor interaction	3 of 217	0.0130

**(C) Reactome Pathways**	**Pathway Description**	**Count in Gene Set**	**False Discovery Rate**
RNO-68877	Mitotic prometaphase	4 of 168	0.00063
RNO-5663220	RHO GTPases activate formins	3 of 116	0.0013
RNO-2500257	Resolution of sister chromatid cohesion	3 of 100	0.0013
RNO-141444	Amplification of signal from unattached kinetochores *via* a MAD2 inhibitory signal	3 of 80	0.0013
RNO-2467813	Separation of sister chromatids	3 of 149	0.0019

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
