# Peer review of "Toxicity of TiO2 Nanoparticles: Validation of Alternative Models"

_ijms, 2020, doi:10.3390/ijms21144855_

Round 1
Reviewer 1 Report
The study is aimed to evaluate and compare the toxicity of titanium dioxide (TiO2) nanoparticles (NP) in three systems, in vivo exposure, air-liquid interface (ALI) exposure, and in vitro submerged exposure, using transcriptomic profile as readout. While ALI seemed to be a reasonable alternative to the in vitro submerged exposure and in vivo exposure, this study does not generate convincing data to either support or oppose the use of ALI in place of in vitro submerged exposure and in vivo exposure in evaluating TiO2 NP toxicity. Below are my comments.
1. Lung samples consist of difference cell types which can respond to TiO2 NP exposure and/or interact with macrophages, therefore, one can not compare the transcriptomic profile of a piece of lung tissue with macrophages cultured in vitro. In fact, the result of this study seemed to be conflicting. One the one hand, “five common biological process (GO-BP) were found between in vivo and ALI exposures, only one between in vivo and in vitro exposures, and none between ALI and in vitro exposures”, while on the other hand, “among the 18 DEGs, 14 DEGs had the same dysregulation in in vitro and ALI conditions while contrary dysregulated in vivo condition; 1 DEG (Ptpn13) was down- regulated in vivo and ALI conditions while up-regulated in vitro submerged conditions”. Overall, it is inconclusive which system, in vitro exposure or ALI, better reflects the effect of TiO2 NP exposure in vivo.
2. In both in vitro and ALI, the behavior of the TiO2 NP seems somewhat unpredictable, so does their effect on macrophages. A better characterization of the system is needed. For example, how much do they form agglomeration, how fast do they sediment, and how well are they taken up by macrophages in each system?
Author Response
Reviewer 1
The study is aimed to evaluate and compare the toxicity of titanium dioxide (TiO2) nanoparticles (NP) in three systems, in vivo exposure, air-liquid interface (ALI) exposure, and in vitro submerged exposure, using transcriptomic profile as readout. While ALI seemed to be a reasonable alternative to the in vitro submerged exposure and in vivo exposure, this study does not generate convincing data to either support or oppose the use of ALI in place of in vitro submerged exposure and in vivo exposure in evaluating TiO2 NP toxicity. Below are my comments.
- Lung samples consist of difference cell types which can respond to TiO2 NP exposure and/or interact with macrophages, therefore, one can not compare the transcriptomic profile of a piece of lung tissue with macrophages cultured in vitro. In fact, the result of this study seemed to be conflicting. One the one hand, “five common biological process (GO-BP) were found between in vivo and ALI exposures, only one between in vivo and in vitro exposures, and none between ALI and in vitro exposures”, while on the other hand, “among the 18 DEGs, 14 DEGs had the same dysregulation in in vitro and ALI conditions while contrary dysregulated in vivo condition; 1 DEG (Ptpn13) was down- regulated in vivo and ALI conditions while up-regulated in vitro submerged conditions”. Overall, it is inconclusive which system, in vitro exposure or ALI, better reflects the effect of TiO2 NP exposure in vivo.
Yes, we know well that in vitro systems could not recreate the specificities and the complexity of an entire tissue. Indeed, lung is composed of 40 different cell types. Among, them resident macrophages, used in that present study, constitute the first immune barrier and are the first cells in contact with the toxicant. The analysis has been made at two levels: the gene clusters level (Biological process) and the gene level. Thus, to validate alternative models, we identified common biological processes or dysregulated genes into the three systems. It was not the purpose of this work to state which system is the better, but to validate their use. Therefore, we identified 4 genes, namely Ccl4, Osm, Ccl7 and Bcl3, dysregulated in the three systems. It means that these genes can be used in submerged or in ALI experiments, when one wants to work on TiO2 NP.
Concerning the analysis of the Biological Processes, we clearly stated “Five common biological process (GO-BP) were found between in vivo and ALI exposures (…) The 5 common GO-BP are involved in metabolic and cellular processes, suggesting that cellular responses in ALI conditions are similar to those obtained in in vivo condition.” As only one BP was identified between submerged and in vivo conditions, ALI better reflects the effect of TiO2 NP exposure in vivo. We confess that it was not clear enough. So, the following sentence was added in the conclusion (lines 498-500): “This analysis permits us to identify five common GO-BP between in vivo and ALI exposures, and only one between submerged and in vivo suggesting that ALI better reflects the effect of TiO2 NP exposure in vivo.”
- In both in vitro and ALI, the behavior of the TiO2 NP seems somewhat unpredictable, so does their effect on macrophages. A better characterization of the system is needed. For example, how much do they form agglomeration, how fast do they sediment, and how well are they taken up by macrophages in each system?
Concerning the agglomeration of TiO2 NP, as showed in Table 1, the primary size of NP (provider information) was 21.5 ± 7.2 nm, and the secondary size in medium, measured by DLS, was 170 ± 1.5 nm. As the DLS was done just after sonication, in the treatment medium (DMEM-SVF free medium), we can consider that the size of NP agglomerates was 170 ± 1.5 nm.
Our ALI system (Vitrocell® Cloud) is coupled with a high precision QCM balance which measures the mass deposition in a well in the system. The nebulization of NM105 resulted in a deposition of 5.39 ± 0.86 µg/cm², confirming the good sedimentation of NP on the cells in the ALI system in 17 ± 3.37 minutes (these results are to be published in an upcoming article). Since the BET specific surface area of this nanomaterial was 51 m2/g, 5 µg/cm² corresponds to a dose exposure of 3 cm² NP/cm² of cells, the same dose exposure that was used in vitro, and observed in vivo.
The sedimentation in medium is a current question asked in nanotoxicology studies. It is well known that sedimentation depends on the medium characteristics as viscosity, temperature, and pH, as well as on the particle properties as size, structure of the primary particle, agglomeration, shape and density. The sedimentation is governed by Stokes-Einstein laws and the Brownian motion laws. The OCDE (Organisation for Economic Co-operation and Development) reports that “while for small particles the sedimentation in the aqueous medium is countered by the Brownian displacement and net sedimentation rate is negligible, the formed agglomerates eventually reach particle sizes at which the sedimentation rate is greater than the Brownian displacement causing the particles to sediment: the dispersion is not stable over time” (OECD, 2017).
Very few articles were published about the NM105 sedimentation. NM-105 sedimentation is most of the time reported after 6 h, but we can consider that between 60 and 80% of NP sediment in 4 hours (OECD, 2017; Tavares et al., 2014). The ISDD computational model, predicted a deposition of 20% to 60% of NP with a size range from 20 to 200 nm, at 10 μg/mL, during 4 h, in a media height of 3.1mm, which corresponds to our in vitro conditions (4 h exposure, 100µL in a 96-well plate, but with 20µg/cm²) (Hinderliter et al., 2010). In addition, we must consider that NR8383 cells are semi-adherent cells, meaning that 50% of cells are in suspension, so we can estimate that the interaction between cells and NP after 4hours are close to the optimum.
Finally, the uptake of NM105 by NR8383 was not verified in our study, but the uptake of nanoparticles by NR8383 was assessed for TiO2 NM105 by Wiemann et al. (2016) after a 16 h exposure at 90 µg/cm², and for ultrafine TiO2 similar to NM105 (80% anatase , 20% rutile, primary size of 25nm) by Scherbart et al. (2011) after a 4- and 24-h exposure at 10µg/cm². We have validated the internalization by NR8383 of polymeric NP and metallic NP in already published work (Eidi et al 2012, Doumandji et al. 2019), meaning that there is no doubt on the internalization of NM105 in that present work.
REFERENCES
Doumandji, Z., Safar; R., Lovera-Leroux; M., Nahle; S., Cassidy; H., Matallanas; D., Rihn; B., Ferrari; L., Joubert, O. (2019) Protein and Lipid Homeostasis Altered in Rat Macrophages After Exposure to Metallic Oxide Nanoparticles. Cell Biol Toxicol . 36(1):65-82.
Eidi, H., Joubert, O., Némos, C., Grandemange, S., Mograbi, B., Foliguet, B., Tournebize, T., Maincent, P., Le Faou, A., Aboukhamis, I., Rihn, B.H. (2012). Drug Delivery by Polymeric Nanoparticles Induces Autophagy in Macrophages. Int J Pharm. 2012 Jan 17;422(1-2):495-503.
Hinderliter, P.M., Minard, K.R., Orr, G., Chrisler, W.B., Thrall, B.D., Pounds, J.G., and Teeguarden, J.G. (2010). ISDD: A computational model of particle sedimentation, diffusion and target cell dosimetry for in vitro toxicity studies. Part Fibre Toxicol 7, 36.
OECD (2017). Test No. 318: Dispersion Stability of Nanomaterials in Simulated Environmental Media (OECD).
Scherbart, A.M., Langer, J., Bushmelev, A., van Berlo, D., Haberzettl, P., van Schooten, F.-J., Schmidt, A.M., Rose, C.R., Schins, R.P., and Albrecht, C. (2011). Contrasting macrophage activation by fine and ultrafine titanium dioxide particles is associated with different uptake mechanisms. Part Fibre Toxicol 8, 31.
Tavares, A.M., Louro, H., Antunes, S., Quarré, S., Simar, S., De Temmerman, P.-J., Verleysen, E., Mast, J., Jensen, K.A., Norppa, H., et al. (2014). Genotoxicity evaluation of nanosized titanium dioxide, synthetic amorphous silica and multi-walled carbon nanotubes in human lymphocytes. Toxicology in Vitro 28, 60–69.
Wiemann, M., Vennemann, A., Sauer, U.G., Wiench, K., Ma-Hock, L., and Landsiedel, R. (2016). An in vitro alveolar macrophage assay for predicting the short-term inhalation toxicity of nanomaterials. J Nanobiotechnol 14, 16.

Reviewer 2 Report
This work investigates the toxicity of TiO2 nanoparticles. It identifies and validates specific markers of lung exposure to TiO2 nanoparticles, by comparing the results of three types of exposure (in vitro, air-liquid interface, and in vivo). The importance of the study is well argued in the light of the wide spectrum of applications of TiO2 nanoparticles.
This manuscript is information-rich, nicely illustrated, but hastily written. The text is verbose and hard to read. The authors are encouraged to perform a thorough polishing of the entire text.
A list of minor errors of style or language is provided below in the format <original text> => <recommended revision>:
Line 19: test => tests
Line 20: I would delete "NM-105" from this line and from line 68. (This acronym is first explained in "Materials and Methods", on line 74).
Line 24: genes => gene
Line 39: 2012), and in the medical domain =>
2012). In the medical domain, TiO2 NP are used
Line 70: , from an ethical point of view, more socially acceptable => ethically more acceptable (Ethics is a social concept.)
Line 93:
Please remove the accent placed accidentally after "reference".
Line 99: variance analysis => analysis of variance
Line 108: entirely made => made
Line 115: the end of rats’ exposure, animal => the exposure period, the animals
Line 117: The expression of the genes => Gene expression
Line 122: in an aerosol form through => in aerosol using
Line 127: Please reduce the font size of the text portion "3 cm2 of ... culture".
Line 130:
The subtitle "Total RNA Extraction" could be placed at the beginning of the paragraph, written in Italic but normal font size, as follows: "Total RNA extraction. In order to ..." Similar formatting would fit for the next two paragraphs.
Line 143: microarrays chips => microarray chips
Line 156: Please reduce the font size of "Venn".
Line 165: is 51 m²/g => was 51 m²/g
Line 170: I would remove Table I because the same information has been described in the last sentence above Fig. 1.
Line 179: Loss of integrity => loss of membrane integrity
Line 180: positive control => positive control (C+)
Lines 185-196, Table II, lines 349-362: Instead of "fold change", I would use the acronym FC defined before (line 150).
Fig. 3: I suppose the volcano plots from this figure will be displayed at proper resolution in the final version of the paper. As they stand, the labeling is not legible even if one tries to magnify the picture.
Line 226: Please reduce the font size of the caption of Fig. 4.
Line 238: Interestingly another => Interestingly, another
Lines 256-259: down regulated => down-regulated
(Another option is to use the simple spelling, "upregulated" and "downregulated".)
Line 275: Unsurprisingly, among these 18 dysregulated genes, several studies and databases showed that they can be => Unsurprisingly, several studies and databases showed that these 18 dysregulated genes are
Line 282: R-RNO-68877) => (R-RNO-68877)
Line 287: distinguish colors => different colors
detailed informations where => detailed information was
Line 315: alveolar macrophages precursors => alveolar macrophage precursors
Line 323: classic in vitro submerged exposition method do not take => typical in vitro submerged exposition methods do not take
Line 326: the surfactant play => the surfactant plays
Line 328: exposed with a nebulizer to first the surfactant and then TiO2 => exposed with a nebulizer first to the surfactant and then to TiO2
Line 330: Please do not format "The" in Italic.
Line 391: and finally electronic microscopy => and electron microscopy
Line 456: Consistently with our precedent observations => Consistently with our previous observations
Line 464: functionally phenotypes => functional phenotypes
Line 468: by creating a tumor microenvironment => by contributing to the tumor microenvironment
Also, a reference is needed here.
Line 497: As a conclusion, Ccl4, Osm, Ccl7 and Bcl3, could be suggested as => This work suggests that Ccl4, Osm, Ccl7, and Bcl3 are
Line 503: regardless ethical => regardless of ethical
Author Response
Thank you for the English corrections, permitting us to enhance the quality of our manuscript. The manuscript has been already corrected by a native English speaker, but all your suggested corrections have been made.

Round 2
Reviewer 1 Report
"Regarding the differences between the fold changes in in vitro submerged experiments, the results comparison was performed using a cut-off of 2.8 for in vitro submerged and 1.3 for in vivo and in vitro ALI conditions. These cut-offs allow a better selective analysis for the comparison of the three exposures."
It is clearly how exactly the cut-off is set. Could the authors please provide a better justification? Please also provide the specific fold change for Ccl4, Osm, Ccl7 and Bcl3 in each system.
Author Response
We thank the reviewer for this question and remark, which will improve our manuscript.
In a preliminary analysis, we have compared the number of DEG in the three conditions with different fold changes, which were reported in table II (copy below): with a cut-off fold change (FC) of 1.3. 11503 DEG were found in vitro, 1053 DEG in ALI, and 1401 in in vivo conditions; with a FC 1.5, the number of deregulated genes in vitro is 7895 DEG, 1721 DEG with a FC 3, and 1905 DEG with a FC 2.8 (Figure 4, copy below).
We have thought more statistically rigorous to compare a similar number of genes in the three conditions. Thus, the stringency was increased to FC 2.8 for in vitro conditions and diminished to FC 1.3 for in vivo and ALI conditions, with a p-value of p<0,05 for the three conditions. From a biological point of view, a FC 1.3 means a 30% differential expression, which is not negligible. As discussed in the manuscript, the differences of DEG number between in vitro, and in vivo and ALI conditions can be explained by the experimental conditions, the suspension of nanoparticles in liquid (vs. in droplets), the presence of surfactant in ALI conditions, or the tissue response in in vivo conditions.
The specific fold-changes are clearly shown in the Table IV (below a copy of the table for these four genes), these four genes are up-regulated in the three conditions.
Please also provide the specific fold change for Ccl4, Osm, Ccl7 and Bcl3 in each system.
Thank you for your request, as suggested, the specific FC for Ccl4, Osm, Ccl7 and Bcl3 in each system were added in the text, and the sentence was corrected in the manuscript as follows:
Line 252-257: “Among the 18 common DEGs identified with the Venn Diagram (Fig. 4), four are upregulated in the three conditions: Ccl4 (FC: 1,71; 3,72; 2,15 for in vivo, in vitro, and ALI conditions, respectively), Ccl7 (FC: 5,61; 3,53; 1,47), and Osm (FC: 1,38; 6,84; 1,41) which are involved in the same KEGG pathways (cytokine-cytokine receptor interaction) and Bcl3 (FC: 1,69; 4,60; 1,41) which plays a role in cell proliferation (Tables IV and V(B)), while the other common DEG have an heterogeneous dysregulation.”
To have a better comprehension of the results and of the choice of cut-off the paragraph was modified in the manuscript, as follows:
Line 219-223: “Regarding the differences between the fold changes in in vitro submerged experiments, the results comparison was performed using a cut-off of 2.8 for in vitro submerged and 1.3 for in vivo and in vitro ALI conditions with a p-value fixed to p<0,05 for the three conditions. These cut-offs focus on the most deregulated genes in the three conditions and allow a better selective analysis for the comparison of the three exposures.”
Other modification:
Table 4, title: “more” was suppressed, and “genestring” was replaced by “string”
